# The Role of the Prognostic Inflammatory and Nutritional Index (PINI) in the Evolution of Patients with Chronic Kidney Disease and Other Pathologies

**DOI:** 10.3390/healthcare11101375

**Published:** 2023-05-10

**Authors:** Monica Cordos, Cristiana-Elena Vlad, Simona-Mihaela Hogas, Roxana Filip, Gabriela Geletu, Maria Bogdan, Codruta Badescu, Ancuta Goriuc, Liliana Georgeta Foia

**Affiliations:** 1Faculty of Medicine, “Grigore T. Popa” University of Medicine and Pharmacy, 700115 Iasi, Romania; 2Faculty of Medicine and Biological Sciences, Stefan cel Mare University of Suceava, 720229 Suceava, Romania; 3Suceava Emergency County Hospital, 720224 Suceava, Romania; 4Faculty of Dentistry, “Grigore T. Popa” University of Medicine and Pharmacy of Iasi, 700115 Iasi, Romania; 5Faculty of Pharmacy, University of Medicine and Pharmacy, 200349 Craiova, Romania

**Keywords:** Prognostic Inflammatory and Nutritional Index, malnutrition, inflammatory, hemodialysis (HD) patients

## Abstract

Background: Protein-energy loss and inflammation are the main risk factors in the occurrence of complications in hemodialysis patients. The Prognostic Inflammatory and Nutritional Index (PINI) is a simple, inexpensive test to identify the early onset of inflammation and malnutrition in hemodialysis patients, critically ill subjects and those with malignancies. Methods: A systemic review of English literature was conducted on the topic published between 1985 and 2022. A focused and sensitive search strategy was applied to the PUBMED database to identify relevant scientific articles in English. Once articles were identified, a detailed quality and bias assessment was performed. Two independent researchers analyzed the detailed data extraction. Results: PINI proved to be a sensitive, powerful, low-cost and simple test. PINI has been useful in assessing evolution and prognostics in clinical care, with values above one being associated with a high risk of mortality and morbidity. It is useful in cases with surgical and postoperative complications, long hospitalization, as well as increased associated expenses. Conclusions: This is the first review of the literature on the above-mentioned topic (PINI) and is a valuable candidate for validating prognosis in patients with different pathologies.

## 1. Introduction

A relationship was observed between nutritional factors and inflammation [1,2], with optimal nutritional status being essential to modulate inflammatory processes. Furthermore, both are related to the immune system [3]. Malnutrition is due to inadequate intake and absorption, which causes a decrease in lean muscle mass and body cell mass, leading to disease [4]. Malnutrition was proved to be an important risk factor for adverse outcomes such as a higher risk of readmission within 30 days, prolonged hospital/ICU stays, increased infection rates and increased mortality [5]. To identify malnutrition, the Global Leadership Initiative on Malnutrition (GLIM) recommends a wide range of tools, including NRS-2002, SGA, MUST or MNA-SF, because there is no universal method for nutritional risk screening [5]. The pathologies commonly encountered in patients with chronic malnutrition associated with inflammation include neoplasia, chronic obstructive pulmonary disease (COPD), inflammatory bowel diseases, congestive heart failure, chronic kidney disease, and other end-stage organ diseases [4,5].

Protein-energy wasting and inflammation represent the main risks for complications following hemodialysis. A simple and inexpensive test for regular screening of patients with maintenance hemodialysis (MHD) has long been sought to detect the early onset of inflammation and malnutrition, as they are major causes of death among hemodialysis patients.

The Prognostic Inflammatory and Nutritional Index (PINI) was developed in 1985 by Ingenbleek and Carpentier as a clinical assessment tool for patients suffering from infections who arrived at the hospital in critical condition [6,7,8]. The authors suggested the use of a laboratory-based nutrition screening process, in which only four markers indicative of the nutritional and inflammatory status were combined. In critically ill hemodialyzed patients, PINI proved to be a sensitive and specific marker for the simultaneous detection of early changes in their nutritional and inflammatory status [9]. PINI was calculated by using the formula: (C-reactive protein × α1-acid glycoprotein)/(albumin × transthyretin) [7,9]. 

This index represents a simple clinical assessment tool that combines two blood markers of inflammation, C-reactive protein (CRP) and α1-acid glycoprotein (also called orosomucoid), with two biomarkers that reflect the nutritional status, namely albumin and prealbumin (or transthyretin) [10,11,12]. The resulting value for PINI is a number value, with 1 representing the established reference value. The normal value should be PINI < 1 (corresponding to the non-inflammatory status), whereas PINI > 1 indicates a major risk of organic complication. It is suggested that major inflammatory process-related changes observed in critically ill patients should be considered when attempting to develop scores for predicting their morbidity and mortality [8]. The PINI was originally derived from several proteins studied in healthy volunteers and by monitoring three groups of patients: ambulatory outpatients, moderately ill patients, and critically ill patients [8]. Because albumin has a large sample size for individuals of all health classes and a much longer half-life (14–20 days), it is not indicative of the immediate nutritional status and may be skewed by changes in hydration status. Serum albumin concentration may be affected by albumin infusion, dehydration, sepsis, trauma and liver disease, remaining independent of the nutritional status [10], yet it decreases in response to inflammation [13]. Albumin is the most abundant protein in plasma or serum [14,15], a valued biomarker of many conditions, which finds clinical application in the therapy of several pathologies, including hemodialysis, hemorrhage, acute respiratory distress syndrome, acute liver failure, chronic liver disease, shock and hypoalbuminemia [15]. Compared to albumin, prealbumin is more sensitive to changes in protein-energy status. Rather than overall nutritional status, prealbumin concentration reflects recent dietary intake [16].

Albumin and prealbumin are known as negative acute-phase proteins, while CRP and orosomucoid are positive acute-phase proteins [10]. Alpha1-acid glycoprotein (orosomucoid) is an inflammatory protein specific to the major acute phase, synthesized by the microvascular endothelium; its presence in the urine possibly indicates underlying inflammation in hemodialysis patients [8]. Orosomucoid is synthesized in the microvascular endothelium of the liver and exerts control by reducing the effect of the inflammatory cascade, thereby protecting the tissues against damage from inflammation [17]. 

CRP is synthesized in the liver, has a half-life of 19 h, and its measurement is widely used as a representative of both acute and chronic inflammatory disease biomarkers. CRP concentrations higher than 2.0 mg/L can be used to indicate inflammatory pathologies [18,19]. CRP concentration may increase more than 1000-fold in severe inflammatory status [20]. 

However, the available data clearly indicate that the PINI system fulfills its most promising potential in identifying subclinical or marginal cases as well as complex inflammatory and nutritional disorders in protracted illnesses [8]. Given that the PINI index is not used on a regular basis, but rather in clinical trials, its widespread use would significantly reduce hospitalization costs in hemodialysis patients. Our aim was to identify prospective trials, retrospective and observational studies assessing the efficiency of the Prognostic Inflammatory and Nutritional Index (PINI) in clinical care.

## 2. Materials and Methods

### 2.1. Search Strategy and Sources

We searched PUBMED with reference to the period between 1985 and 2022. Two of the researchers (CM and CEV) were independently responsible for screening the identified articles for eligibility.

### 2.2. Inclusion and Exclusion Criteria

Articles were screened by abstract and then by full text about inflammatory and nutritional status of hemodialyzed patients; all the analyzed studies were written in English. We only included studies in which the research population was aged. We included those studies that compared several analyzed indexes: albumin, prealbumin, orosomucoid and CRP, and we excluded those that did not calculate the PINI index, as proposed in the studies conducted by Ingenbleek and Carpentier. Search terms: “PINI”, “inflammatory status”, “nutritional status”, “orosomucoid”, “CRP”, “albumin”, “hemodialysis”, “oncology”, “chronic disease”.

### 2.3. Date Abstraction

From the eligible studies, we collected information on study characteristics (CRP, orosomucoid, albumin, prealbumin) and their connection with PINI calculated as (C reactive protein × α1-acid-glicoprotein)/(albumin × transthyretin), along with evidence on the prognosis of inflammatory status correlated with PINI values.

## 3. Results

### 3.1. Search Results

Our search of the literature indicated 45 published articles. The analysis of their titles and abstracts reduced the potential number of eligible studies to 27. After applying the exclusion criteria, we identified 13 publications involving a total of 1114 subjects to be used for this review. Nine of these studies were prospective trials, two were retrospective and two were observational studies (Figure 1).

### 3.2. Study Characteristics

Table 1 presents details on the analyzed studies: author, year of publication, study design, sample size, mean age, type and duration of intervention, PINI measures and conclusion. The studies were conducted in various locations throughout the world, including the United States of America (USA), European countries and Africa.

### 3.3. Association with Outcomes

Due to the diversity of studies, we divided them according to the pathology: four studies in critical patients—intensive care unit (ICU), four studies in hemodialysis patients, three studies on advanced neoplasia and two on chronic diseases. The studies were carried out for the following periods: 1 month in two studies, 6 months in two studies, 2 months in one study, 32 months in one study, 70 months in one study, 3 months in one study, 7 months in one study and no time period recordings for four of our studies. Most of the studies involved a small number of patients, with the smallest sample size including 15 subjects [21], while the largest included 231 participants [17]. All of the authors used the same formula to calculate PINI (C reactive protein × α1-acid-licoprotein)/(albumin × transthyretin) [22]. 

#### 3.3.1. The PINI Score in Intensive Care Unit (ICU)

The median PINI value was in approximately the same range as the prospective studies conducted by Schlossmacher and Vehe et al. [22,23], whereas the PINI score decreased significantly (by up to 85%) compared to the baseline value reported in the study conducted by Vehe et al. [23].

The PINI value at the extubation moment predicted the subsequent outcome, particularly the likelihood of reintubation. A reduction in PINI levels was associated with independently successful extubation (OR = 7.25, 95% CI 0.04–0.77, *p* = 0.015) and a decreased risk of death (OR = 0.11, 95% CI 0.02–0.67, *p* = 0.012) [22]. 

In the prospective study, Vehe et al. came to the conclusion that increased values of the PINI score were correlated with elevated values of CRP and the APACHE II score, with low values of prealbumin, low energy intake and decreased protein intake [23]. 

Reynold et al. found a statistically significant difference in PINI values between patients whose pressure ulcers improved and those whose ulcers worsened [21].

The PINI score is useful for improving pressure ulcer risk estimation and for identifying scarce equipment in order to prevent their occurrence [21]. Furthermore, Gharsallah et al. found that an increased PINI score was not associated with all-cause mortality. However, they identified an initial increase followed by a progressive decrease in CRP and orosomucoid values and an increase in serum prealbumin and albumin values [10]. 

#### 3.3.2. The PINI Score in Hemodialysis (HD) Patients

The PINI values ranged between 0.67 ± 0.23 and were not significantly modified throughout the duration of the study that enrolled 18 HD patients. Serum albumin and prealbumin concentrations also exhibited a significant increase after the correction of metabolic acidosis [24].

In the cross-sectional study that included 177 HD patients, randomized into two groups (with and without cardiovascular-CV events), Terrier et al. did not observe an association between dialysis modalities (either standard HD or hemodiafiltration) and the proinflammatory status (CRP, α1-acid-glycoprotein, PINI) [25]. The highest tertile of PINI (over 1.56) was associated with an OR of 3.44, following the adjustment for CV risk factors in enrolled patients [25]. After adjustments according to age, sex and dialysis center, the highest tertile of CRP was associated with a significant decrease in nutritional measures (transthyretin, albumin) and HDL cholesterol [25]. 

In a study that included 52 HD patients, divided according to BMI (body mass index), Di Renzo et al. revealed that only 19 HD patients recorded a PINI score over 1 [26]. In non-obese HD subjects, a significant positive correlation was observed between weight/hip ratio and 1-AGP and between lean body mass and prealbumin, while in pre-obese–obese HD subjects, the correlation between pre- and post-dialysis BMI and prealbumin was significantly positive. Pre-obese patients recorded the highest blood levels of the inflammatory pattern [26].

The study conducted by Dessi et al. included 121 adult dialysis patients, monitored for 32 months and examined every 6 months for PINI, subsequently revealing that the survival rate of patients with a PINI score above 1 was significantly lower than that observed in patients with albumin low serum levels (*p* < 0.05) or elevated CRP (*p* < 0.05). In addition, HD patients with PINI > 1 had at least one cardiovascular (CV) event compared to patients with PINI < 1. At the same time, eight end-stage renal disease (ESRD) patients with a PINI value < 1 underwent successful transplants [9].

#### 3.3.3. The PINI Score in Solid and Haematological Malignancies

Costa et al. performed an analysis of longitudinal, descriptive and analytical studies of patients with gastrointestinal cancer and candidates for elective surgery and found that a PINI score over 1, Glasgow score over 2, high subjective global assessment of nutritional status (SGA), low serum albumin and high CRP have been associated with high incidences of complications (e.g., malnutrition) and a high death rate [7].

In a study that enrolled 50 patients with oncological pathologies, the mean PINI score was 102 (SD 142, 95% CI 62–142), with no statistically significant differences following the administration of corticosteroids. Furthermore, no correlation was found between weight loss, age, gender and PINI [27]. 

The PINI score was determined in 231 patients with multiple myeloma without previous treatment. In the overall population and the elderly subgroup (persons aged 65 years or older), PINI > 4 (high PINI) was associated with shorter median survival, 26 versus 65 months, in the PINI < 4 group [29,30]. The prognosis offered by the PINI index was dramatic in the elderly multiple myeloma (MM) subgroup. High PINI predicted shorter survival rates in various groups with good prognosis, such as low International Staging System (ISS) stages and absence of del17p and t(4;14), further validating its prognostic impact on overall survival [28].

#### 3.3.4. The PINI Score in Chronic Diseases

In an observational study, Lenartova et al. revealed that PINI in the group of patients with COPD presented an average value of 4.65 ± 10.77 and 0.026 ± 0.025 in smoking patients. Thus, inflammation and the catabolic process are frequently encountered in smoking patients with COPD [17].

## 4. Discussion

Our review targeted the study of PINI usage in clinical care with the aim of pointing to the importance of this index in various pathologies [28]. To our knowledge, this is the first systematic review of the literature on the applicability of PINI in clinical care. All included scientific materials were published in English as inclusion criteria, which might impact the final number of evaluated articles. Studied articles were from different continents (Europe, North America, South America and Africa), with most of the articles published in France, followed by the USA and Italy. The PINI reveals a significant heterogeneity regarding the different interventions tested (various combinations of markers), also in terms of sample size and duration.

Clinical heterogeneity cannot be translated into statistical heterogeneity; hence, we explore heterogeneity by analyzing subgroups. When comparing the studies according to their type, they can be divided into studies on acute pathology: ICU (intensive care unit) condition, neoplasia with surgery and chronic disease (hemodialysis, multiple myeloma, kidney disease, COPD (chronic obstructive pulmonary disease)). 

Nutritional therapy is important in the treatment plan of patients with malnutrition in association with biological parameters for the improvement and individualization of nutritional management [31]. Several blood biomarkers, including albumin, prealbumin, hemoglobin, total cholesterol and total protein, are indicators of malnutrition, even in the presence of chronic inflammation [32].

As the included studies evaluated a wide range of disorders and were of different duration, a meta-analysis was not feasible. PINI values over 1 mean malnutrition and inflammation; however, without a maximum limit, it can reach very high values in the case of neoplasia and those in the ICU. PINI values are close to 0 but increase progressively in subjects suffering minor stresses, further increasing in critically ill patients in proportion to their nutritional deficits and/or superimposed inflammatory burden [33]. Even if the same formula for PINI was used, the results were different. This aspect can be explained by the wide range of diseases considered. The PINI index was used to encompass the two significant aspects of a chronic condition (nutritional and inflammatory) [24]. In acute events, CRP is the most important determinant of the PINI score due to the amplitude of the response, while in chronic inflammation, the relative intake of visceral proteins (albumin and transthyretin) and acute phase proteins (CRP and acid glycoprotein a1) is more balanced [25]. Evidencing the factors that determine the initiation and progression of inflammation is important for establishing therapeutic targets [1]. Malnutrition affects the immune response by reducing the regeneration and functioning of immune cells (decreased bactericidal function of neutrophils, the complement system, IgA, as well as antibody response), causing more infections [3,34]. Albumin and prealbumin are the proteins most widely studied for diagnosis of malnutrition, being useful indicators for general nutritional status [32].

The smallest sample size was represented by a group of 15 patients [23], while the largest included 231 patients [28]. It is worth pointing out that all the studies described an inadequate follow-up period, with two notable exceptions (Dessi et al. [9] and Dupire et al. [28]), while four of the studies did not display bias for PINI [10,25,26,28]. All 13 studies proved that changes in PINI score over time are more informative and reliable than the absolute values of the parameters included in the formula. In addition, the fact that only two of the included studies used mortality-related outcomes was quite unsatisfactory [9,28].

With regard to studies on patients in critical condition (ICU), the main problem identified was the short follow-up period. Furthermore, due to the sample size, limitations of the time effects of PINI could also be considered. A decrease in PINI levels was associated with a significant reduction in the risk of death [7,21,27,28]. Malnutrition has long been recognized as a risk factor for postoperative morbidity and mortality. In patients with pressure ulcers, PINI appears to offer a potential indicator of prognosis for those patients whose ulcers are likely to get worse [21]. In ICU patients, the increase in PINI was correlated with the early appearance of stress ulcers, multiple organ failure and increased all-cause mortality [7,21,27,28]. Furthermore, PINI > 1 was associated with various postoperative complications, being an essential element in establishing the prognosis [17,22,23,26]. 

CKD patients have a high risk of malnutrition, causing increased morbidity and mortality. The catabolic state is mediated by acidosis, acting in part by increasing the oxidation of branched-chain amino acids and the secretion of catabolic hormones, promoting the synthesis of proteolytic enzymes. Metabolic acidosis triggers increased catabolism of endogenous proteins [24]. 

In hemodialysis patients, BMI is a marker of body composition, their weight being determined by the overhydration status. Malnutrition was identified in patients with a high PINI score (over 1) [9,29]. Low BMI is an essential element for the diagnosis of malnutrition, but its lower limit (BMI of 18.5 kg/m^2^ according to the WHO) cannot identify patients at risk of malnutrition [32,35]. Hemodialysis patients with increased PINI have a clinically silent proinflammatory status (malnutrition-inflammation complex syndrome—MICS) because different infectious and inflammatory diseases remain asymptomatic for a long time due to chronic immunosuppression [9,29]. The PINI is a simple marker to measure inflammatory disease and could be used as a first-line test to identify patients at higher risk of MICS [9]. Chronic inflammation and malnutrition can aggravate the evolution of hemodialysis patients through decompensated heart failure, the occurrence of atherosclerotic cardiovascular disease and increased susceptibility to infections. These two diseases are parts of MICS and must be detected early by using a sensitive, powerful, low-cost and simple test, represented by PINI, being a candidate for implementing a routine test for hemodialysis subjects [9]. 

In HD patients, the increased value of PINI (>1) was associated with the occurrence of atherosclerosis, cardiovascular events and increased cardiovascular mortality [8,9,10,29]. Inflammatory reactions represented by vascular access infection, biocompatibility of dialysis procedures, back-filtration of nonsterile dialysate and periodontal disease can initially influence CRP with a subsequent affliction of the muscle mass and nutritional indices [25]. Inflammation is one of the main determinants of protein-energy malnutrition. Clear interrelationships were observed between CRP and visceral proteins, such as albumin and transthyretin, suggesting that inflammation might be a key element in the development of atherosclerotic CVD in hemodialysis patients [25].

Furthermore, changes in visceral proteins and inflammatory syndrome are associated with impairment of cholesterol levels, supporting the idea that “reverse epidemiology” may be a consequence of the malnutrition–inflammation complex syndrome [25]. The synchronous occurrence of diabetes and arterial hypertension contributes to the development of chronic kidney disease in obese people. In addition, high BMI causes glomerular hyperfiltration, which can independently lead to chronic kidney disease. In hemodialysis patients, the impact of BMI may be obscured by other pathological impairments, including malnutrition and inflammation [26].

The PINI score is an important marker for monitoring the elderly at risk of severe complications occurring in the clinical stage of the disease and as a marker of mortality in patients in the terminal stages of the disease [17]. Also, the adipose tissue, as an endocrine organ, is responsible for the inflammatory status, increasing the risk of obesity-related diseases such as CVD [25].

In the case of patients with solid and haematological malignancies, the increase in the PINI score determined a reduction in the survival rate [7,27,28]. The impact of the PINI index on treatment-related toxicity was studied in patients with solid neoplasia. The high PINI index indicated a good correlation with the risk of severe hematologic toxicity [28]. The high PINI has been used as a predictor of the risk of infection in burn patients and the risk of complications in cancer patients [7]. In oncological pathology, PINI over 1 is strongly related to surgical complications, longer hospitalization and risk of death [22,23]. Moreover, the PINI index could predict side effects in the development of new agents in multiple myeloma [28]. The largest study duration was 70 months; it included patients with multiple myeloma and revealed the use and impact of PINI in assessing the evolution, risk of recurrence and mortality.

The 2020 Updated Clinical Practice Guideline for Nutrition in CKD recommended the use of specific scores to assess nutritional status in patients with end-stage renal disease [36]. In 2018, the European Society for Clinical Nutrition and Metabolism (ESPEN) proposed a universal score to standardize the definition of malnutrition. The Global Leadership Initiative on Malnutrition (GLIM) was developed from a two-step approach: first, a screening to select patients at risk and, subsequently, an assessment to diagnose and grade the severity of malnutrition [37]. However, these are markers of nutrition rather than markers of inflammation. There is no agreement on the optimal method of screening for nutritional risk and assessing nutritional status; different measures, equipment and formulas lead to varying results at both individual and population levels. All this leads to a lack of agreement regarding the definition and evaluation of nutritional status [38]. The 7p-SGA and MIS are tools to evaluate the nutritional status and are composed of nutritional history (changes in body weight and dietary intake, presence of gastrointestinal symptoms), functional capacity, disease-related comorbidities and physical examination (loss of subcutaneous fat, muscle wasting, and clinical edema) [39]. MIS adds the objective assessments of BMI, serum albumin and serum transferrin [39]. The 7p-SGA and MIS, on the other hand, are easy-to-perform methods, simple, inexpensive and recommended by the 2020 National Kidney Foundation/Kidney Disease Outcome Quality Initiative Nutrition guidelines for nutritional assessment [39]. The interpretations of these scores demonstrate that there is a significant difference in their measurement and interpretation by doctors with different levels of experience, thus demonstrating the dependence on the experience and skill of the doctors [39]. The subjective global assessment of nutritional status and the malnutrition-inflammation score (MIS) are considered highly reliable prognostic markers of mortality, morbidity and progression of CKD [40]. Although the individual use of nutritional and inflammatory markers, such as cholesterol and albumin, is not recommended, their combined use in prognostic scores has an important predictive value [36,40].

There are several limitations to our review. First of all, we only included published literature, full texts, and thus the result may be influenced by publication bias. Furthermore, the duration of follow-up was short in most trials, and hence, it requires additional research using trials with longer follow-ups and using mortality as a primary endpoint. The limited data on deaths in our trials did not allow us to examine this issue further. Our review also underlines the important clinical, research and health policy implications in assessing such an indicator. These findings advocate for using PINI in assessing the evolution of critical patients and evaluating mortality in chronic disease.

## 5. Conclusions

Our review is the first so far to evaluate the use of the Prognostic Inflammatory and Nutritional Index (PINI) in clinical care. Malnutrition and inflammation were frequent findings in patients with hemodialysis, cancer, acute respiratory failure and chronic disease. It is unequivocal that PINI can contribute to global management in clinical care and, without assertion for sufficient specificity or sensitivity to be routinely employed in clinical practice, it can serve as a suitable marker for monitoring the elderly at risk of severe complications that occur in clinical stage hemodialysis patients. Larger studies are needed to determine the usefulness of PINI in clinical care.

## Figures and Tables

**Figure 1 healthcare-11-01375-f001:**
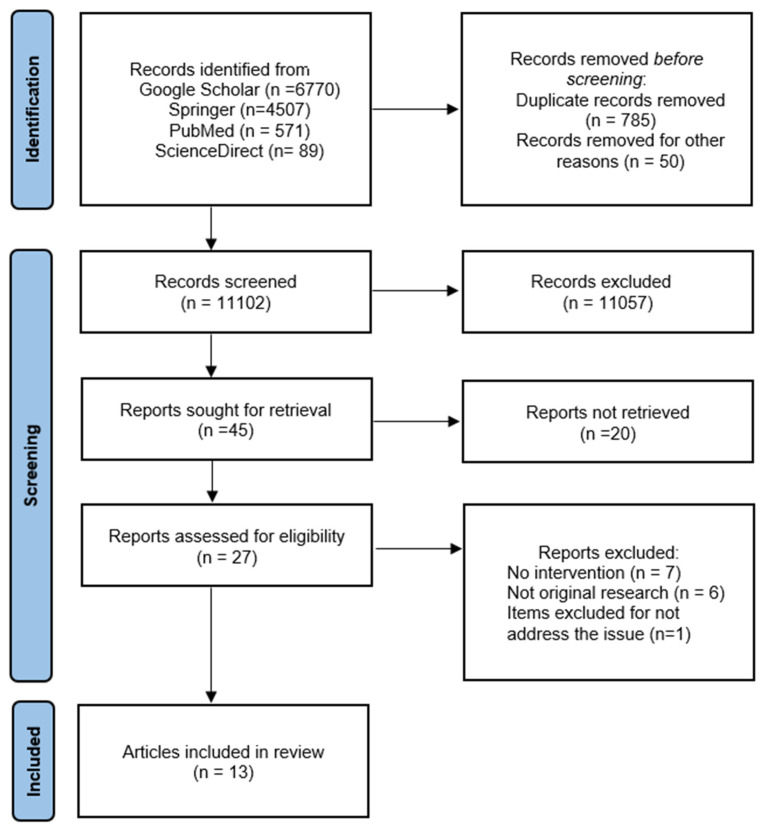
A PRISMA flow diagram for the article selection process in this review.

**Table 1 healthcare-11-01375-t001:** Characteristics of included studies that evaluated the PINI score.

Authors	Country/Location	Study Design	Setting	Follow Up	Sample Size	Mean Age (Years)	Type of Intervention	PINI	Conclusions
1. Costa et al., 2016 [7]	Brazil	Prospective	Gastrointestinal neoplasia	1 month	29	60.6 ± 15.15	Surgery Parenteral nutrition	PINI > 1 (62.1%) PINI < 1 (37.9%)	-PINI > 1 associated with surgical complication and death
2. Dessi et al., 2009 [9]	Italy	Prospective	HD	32 months	121	59.64 ± 14.75	NR	PINI < 1 (71.66%) PINI ≥ 1 (28.33%)	-PINI > 1 associated with higher risk of mortality and morbidity
3. Gharsallah et al., 2014 [10]	Tunis	Prospective	ICU	3 months	20	56 ± 11	Enteral nutrition Mechanical ventilation	PINI > 1 (100%)	-Mortality 35% -PINI correlated with organ failure but not with mortality
4. Lenartova et al., 2017 [17]	Poland	Observational—Prospective	COPD	NR	120	NR	60 patients with COPD 60 patients–control group (healthy individuals)	PINI + COPD > 1 PINI + Control Group < 1	-PINI in COPD is significantly higher than in the control group
5. Reynolds et al., 2006 [21]	UK	Prospective	ICU-pressure ulcers	NR	158	NR	NR	PINI > 1	-PINI higher in patients whose ulcers are likely to worsen
6. Schlossmacher et al., 2002 [22]	France	Prospective	ICU	6 months	83	63.9 ± 15	-Mechanical ventilation -Ant biotherapy -Nutritional support	PINI > 1 Admission: >1 Extubating: >1	-Declining PINI was associated with a reduction in the risk of death
7. Vehe et al., 1991 [23]	USA	Prospective	ICU	28 days	15	NR	Enteral nutrition	PINI > 1	-PINI decreases significantly over time in critically ill traumatized patients
8. Verove et al., 2002 [24]	France	Prospective	HD	6 months	18	73 ± 6	-Sodium bicarbonate oral supplementation	PINI: <1	-No significant change
9. Terrier et al., 2005 [25]	France	Observational—Prospective	HD	NR	177	67.73	CVD+ CVD−	PINI < 1 (71%) PINI > 1 (49%) PINI < 1 (46%) PINI > 1 (11%)	-PINI strongly associated with prevalence of atherosclerosis after adjustment for age, gender, dialysis center
10. Di Renzo et al., 2008 [26]	Italy	Prospective	HD	8 weeks	52	50 ± 11.41	-Non-obese (NO) -Pre-obese (PO) -Acetyl salicylic 100 mg/day, -Atorvastatin 10 mg/day		-Combined treatment was effective in reducing inflammatory status
11. Nelson et al., 2002 [27]	USA	Prospective	Advanced cancer	NR	50	64	NR	PINI > 1	-Significantly elevated PINI scores in patients with advanced cancer
12. Dupire et al., 2012 [28]	France	Retrospective	Multiple myeloma	70 months	231	64	NR	PINI > 1 (36.4%) PINI < 1 (63.6%)	−55% of the patients died -PINI > 4 shorter median -survival (24 ± 19 months) -PINI < 4 median survival (74 ± 5 months) -PINI useful to determine prognosis
13. Kirov et al., 2019 [29]	France	Retrospective	Retroperitoneal liposarcoma	7 months	40	61	Surgery Parenteral nutrition	PINI > 1 (50%) PINI < 1 (50%)	-PINI > 1 associated with longer hospitalization and postoperative complication

Legend: ICU—intensive care unit; HD—hemodialysis; PINI—Prognostic Inflammatory and Nutritional Index; NR—not recorded; COPD—chronic obstructive pulmonary disease; CVD—cardiovascular disease; NO-non-obese; PO-pre-obese; NR—no response.

## Data Availability

Not applicable.

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
