# Peer review of "The Role of the Prognostic Inflammatory and Nutritional Index (PINI) in the Evolution of Patients with Chronic Kidney Disease and Other Pathologies"

_healthcare, 2023, doi:10.3390/healthcare11101375_

Round 1

Reviewer 1 Report

This review aims to discuss the prognostic inflammatory and nutritional index (PINI) in evaluating various pathologies including chronic kidney disease. The authors have searched different databases and obtained 13 filtered papers for such discussion. Although evaluating different pathologies based on a reliable index is an interesting project, the PINI mentioned in this paper is too simple to reflect the real and complex pathologies. The current index only contains four parameters related to nutritional and inflammatory status in patients. In contrast to the simple index, nutrition and inflammation are very broad topics and involve many complex biological processes. To comprehensively reflect the pathologies, authors can involve other potential indexes to assist the current index that is used to evaluate pathologies. Also, the authors should discuss the unique advantages of the current index compared with other indexes.   

The English language is fine, but the overall layout and content coherence can be improved to attract readers. 

Author Response

We would like to thank the Reviewers for their valuable time and useful contribution. We appreciate greatly their input which definitely helped improve our manuscript.

Also, we look forward to hearing from you regarding our submission. We would be glad to respond to any further questions and comments that you may have.

Please find below a detailed point-by-point reply to the comments made by editors.

Reviewer 2 Report

line 35-36

"was developed in 1985 by Ingenbleek and Carpentier" – please cite formally the relevant item for the references (as [4]).

Figure 1: in the boxes there are included one and two asterisks signs ("Records identified from*" and "Records excluded**"). Where are the references to these asterisks in the text?

line 46

"The resulting value for PINI is numerical" – I think it's better to write. "The resulting value for PINI is a number value"

In practice the PINI exactly equal to 1 is a very rare case. Does this value appear in practice? Please also specify what values the PINI parameter achieves in practice. Are these values e.g. close to zero or many times greater than 1? What values are found in practice? For example in line 191 you write "There is no universal threshold for PINI index, which varies from 1 in pediatric patients to 25 in geriatric patients [13]". Does that mean he's never over 25? Confronting this information with the data provided in the table in the PINI column, one may have at least considerable doubts

line 46

"as a result of a fraction" – explain what did you mean exactly. Now, the phrase used is incomprehensible.

Table 1

There is no header in the included table (should be "Table I"). Some entries in the PINI column are strange to say the least. How should such records be understood: "PINI DAY 1(186±202) PINI DAY 14(27±40)", "PINI mean: 102" (why only the mean value is presented), "(NO)PINI", "(PO)PINI" etc. After all, it is clearly written that this index can take values <1, 1, >1. So what values are given in the screened articles? Please think carefully about what information to include in this column so that it has some practical usefulness. There is a kind of disorder at the moment.

linia 83

We only included studies in which the research population was aged. What does the term elderly population mean? Please specify e.g. by referring to the WHO classification

line 304

Please remove the quotation mark

Author Response

(The authors gave the same response as above.)

Round 2

Reviewer 1 Report

This manuscript has been improved and could be recommended for publication.